# GelMA Hydrogel Reinforced with 3D Printed PEGT/PBT Scaffolds for Supporting Epigenetically-Activated Human Bone Marrow Stromal Cells for Bone Repair

**DOI:** 10.3390/jfb13020041

**Published:** 2022-04-10

**Authors:** Kenny Man, Cesar Alcala, Naveen V. Mekhileri, Khoon S. Lim, Lin-Hua Jiang, Tim B. F. Woodfield, Xuebin B. Yang

**Affiliations:** 1Biomaterials & Tissue Engineering Group, School of Dentistry, University of Leeds, Leeds LS9 7TF, UK; k.l.man@bham.ac.uk; 2School of Chemical Engineering, University of Birmingham, Edgbaston, Birmingham B15 2TT, UK; 3CReaTE Group, Department of Orthopaedic Surgery, University of Otago Christchurch, Christchurch 8011, New Zealand; cesar.alcala@otago.ac.nz (C.A.); naveenmv.14@gmail.com (N.V.M.); khoon.lim@otago.ac.nz (K.S.L.); tim.woodfield@otago.ac.nz (T.B.F.W.); 4School of Biomedical Sciences, University of Leeds, Leeds LS2 9JT, UK; l.h.jiang@leeds.ac.uk

**Keywords:** HDAC inhibitor, MI192, epigenetics, hydrogel, GelMA, 3D printing, bone, tissue engineering

## Abstract

Epigenetic approaches using the histone deacetylase 2 and 3 inhibitor-MI192 have been reported to accelerate stem cells to form mineralised tissues. Gelatine methacryloyl (GelMA) hydrogels provide a favourable microenvironment to facilitate cell delivery and support tissue formation. However, their application for bone repair is limited due to their low mechanical strength. This study aimed to investigate a GelMA hydrogel reinforced with a 3D printed scaffold to support MI192-induced human bone marrow stromal cells (hBMSCs) for bone formation. Cell culture: The GelMA (5 wt%) hydrogel supported the proliferation of MI192-pre-treated hBMSCs. MI192-pre-treated hBMSCs within the GelMA in osteogenic culture significantly increased alkaline phosphatase activity (*p* ≤ 0.001) compared to control. Histology: The MI192-pre-treated group enhanced osteoblast-related extracellular matrix deposition and mineralisation (*p* ≤ 0.001) compared to control. Mechanical testing: GelMA hydrogels reinforced with 3D printed poly(ethylene glycol)-terephthalate/poly(butylene terephthalate) (PEGT/PBT) scaffolds exhibited a 1000-fold increase in the compressive modulus compared to the GelMA alone. MI192-pre-treated hBMSCs within the GelMA–PEGT/PBT constructs significantly enhanced extracellular matrix collagen production and mineralisation compared to control (*p* ≤ 0.001). These findings demonstrate that the GelMA–PEGT/PBT construct provides enhanced mechanical strength and facilitates the delivery of epigenetically-activated MSCs for bone augmentation strategies.

## 1. Introduction

The repair of damaged bone arising from traumatic injury or pathological conditions such as osteoporosis, tumours and congenital bone disorder creates a tremendous clinical need, which is anticipated to rise substantially in the future due to our growing ageing population [1,2]. Current gold-standard treatments are associated with numerous drawbacks such as donor site morbidity and limited availability for autografts [3] and immune response resulting in rejection or transmission of disease from allografts [4]. Bone tissue engineering is seen as an alternative approach in meeting the ever-increasing clinical demand for bone tissues [5]. Mesenchymal stromal cells (MSCs) have been extensively researched for tissue engineering applications due to their ease of procurement from multiple tissues and their multipotency [6]. Although the use of MSC-based treatments has shown promise, their clinical translation is hindered due to uncontrolled differentiation, immune rejection, inherent heterogeneity, functional tissue engraftment and neoplasm formation, and the efficacy of tissue formation [7,8]. Recent technologies such as gene therapy have demonstrated their potential in controlling the lineage-specific differentiation of MSCs. However, issues associated with the intensive costs involved and the risk of tumourigenesis hinder clinical adoption [9]. Hence, there is great precedence to develop novel strategies to promote the clinical efficacy of MSC-based therapies for bone repair.

An increasing body of evidence has demonstrated the critical role that epigenetic regulation plays in controlling cellular functions [10]. Epigenetic approaches have garnered increasing interest in regenerative medicine due to their ability to control the transcriptional potential of cells without altering the genome [11,12]. In particular, hyperacetylation induced by histone deacetylase inhibitors (HDACis) has been shown to promote the osteogenesis of MSCs [13,14,15]. The HDAC2 and 3 selective inhibitor-MI192 has been reported to enhance the osteogenic capacity of stem cells derived from adipose and dental tissues [16,17]. Although the potential of HDACis has been displayed in 2D culture, there have been limited investigations into the therapeutic efficacy of these compounds in a 3D microenvironment.

Hydrogel systems have been extensively investigated as cell culture matrices to promote bone regeneration due to their biocompatibility, high water content and ease of administration [18,19]. Gelatine methacryloyl (GelMA) hydrogels are extremely attractive due to their biocompatible, biodegradable and low-cost nature [20,21,22]. Additionally, the photo-curable nature of GelMA permits hydrogel gelation in situ or for additive manufacturing into biomimetic architectures [23,24,25]. Moreover, GelMA hydrogels have been harnessed as a delivery vehicle for cells due to their hydrated 3D microenvironment that supports cell adhesion and functionality. Although GelMA has been used for numerous tissue engineering applications [26,27,28], its use for bone regeneration is limited due to its inherent lack of mechanical strength required for load-bearing tissues [29]. Increasing GelMA concentration has been reported to improve its mechanical properties [20]. However, this comes at the expense of altering the microstructure of the hydrogel, ultimately augmenting its ability to support bone formation [30]. Hence, there is growing precedence for enhancing the mechanical properties of hydrogel systems without affecting their favourable cell-suitable microenvironment. An increasing number of studies have investigated combining advantageous cellular environments with an external scaffold framework to enhance functionality for load-bearing applications [31,32]. One such example is the bio-assembled microtissue (BMT) model that combines pre-cultured cell pellets within a 3D printed poly(ethylene glycol)-terephthalate/poly(butylene terephthalate) (PEGT/PBT) scaffold framework [31]. Although this model has demonstrated its potential for bone tissue engineering, the use of cell pellets is associated with numerous limitations—pellet fabrication is time-consuming and labour intensive, in addition to requiring a high quantity of MSCs, factors that are further exacerbated in the repair of large bone defects [31]. Thus, GelMA hydrogels reinforced with a 3D printed PEGT/PBT structural scaffold could provide a suitable platform to support HDACi (MI192)-induced human bone marrow stromal cells (hBMSCs) for bone formation.

Initially, we investigated the effect of GelMA hydrogels alone on the viability, proliferation and mineralisation of MI192-pre-treated hBMSCs. Next, we assessed the effect of incorporating the 3D printed PEGT/PBT scaffold on the mechanical strength of GelMA hydrogels. Finally, the GelMA–PEGT/PBT construct was evaluated for its capacity to support the extracellular matrix production and mineralisation of MI192-induced hBMSCs.

## 2. Materials and Methods

### 2.1. Macromer Preparation

Type A porcine skin gelatine (Sigma-Aldrich, Gillingham, UK) was fully dispersed at 5% (*w*/*v*) in Dulbecco’s phosphate-buffered saline (PBS, Lonza, Manchester, UK) at 50 °C. Methacrylic anhydride (Sigma-Aldrich, Gillingham, UK) was then added to the gelatine solution under constant stirring at 50 °C and further incubated for 1 h. The mixture was dialysed against distilled water using 12–14 kDa cut-off dialysis tubing for 2–3 days at 40 °C to remove salts and methacrylic acid. The solution was adjusted to pH 7.4 and then sterile filtered. The solution was stored at −80 °C for 24 h and then freeze-dried for 4 days. Freeze-dried GelMA macromer was fully dispersed in D-PBS containing visible-light initiators (0.2 mM tris (2,2′-bipyridyl) dichlororuthenium (II) hexahydrate (Ru) (Sigma-Aldrich, Gillingham, UK) and 2 mM sodium persulfate (SPS) (Sigma-Aldrich, Gillingham, UK)).

### 2.2. GelMA Hydrogel Synthesis

The photo-initiators were added to the pre-polymer solution, and 60 μL of the solution was transferred into silicon moulds (Ø 5 × 2 mm) placed on a glass slide and exposed under a visible-light source (50 ± 5 mW/dm^2^) for 10 min. Once cross-linked, hydrogels were placed into non-adherent 48-well plates and cultured in basal media.

### 2.3. Three-Dimensional Printing of PEGT/PBT Scaffolds

Poly(ethylene glycol)-terephthalate/poly(butylene terephthalate) (PEGT/PBT) block copolymers were fabricated as previously described [33]. Briefly, PEGT/PBT scaffolds (300 g/mL, 55:45 wt%) were fabricated via fused deposition modelling in a layer-by-layer process using a 3D BioPlotter (EnvisionTec, Gladbeck, Germany) with a 1 mm fibre spacing. Fibres were oriented in a 0–90°–90°–0° pattern in order to provide porosity in the *x*–*y* and *x*–*z* planes. The following 3D printing parameters were applied: (i) 1 mm fibre spacing in *x* and *y* directions, (ii) 0.22 mm fibre height offset, (iii) printing temperature of 200 °C, (iv) dispense head pressure and speed of 5 bar and 63 RPM and (v) *x*–*y* nozzle (25 gauge) speed of 500 mm min^−1^. Additionally, 70% ethanol was used to sterilise the scaffolds (3.3 × 2.1 × 2.1 mm^3^) overnight, followed by washing in plain medium before use. For GelMA–PEGT/PBT constructs, PEGT/PBT scaffolds were inserted into the cell/pre-polymer solution prior to cross-linking as described above.

### 2.4. Cell Culture and MI192 Treatment

Human bone marrow stromal cells (hBMSCs) were procured from Lonza (Manchester, UK) and subsequently cultured at 37 °C in 5% CO_2_ in air in a basal medium. The basal culture media consists of alpha modified minimum essential medium (α-MEM, Lonza, Manchester, UK) containing 10% foetal calf serum (FCS, Sigma-Aldrich, Gillingham, UK) and 100 units/mL penicillin with 100 μg/mL streptomycin (Sigma-Aldrich, Gillingham, UK). Cells were passaged when approaching 80% confluence, and cells at passage 4 were used for this study. At 80% confluence, hBMSCs were incubated in a basal medium supplemented with 50 µM MI192 for 48 h as previously reported [33], before combination with the GelMA hydrogel.

### 2.5. Compressive Testing of GelMA–PEGT/PBT Constructs

Compressive Young’s modulus was calculated from the linear region of the resulting stress–strain curves (15–20% strain) obtained with an MTS Criterion^®^ 42 mechanical testing machine (MTS, Eden Prairie, MN, USA) using a 500 N load cell. Measurements (*n* = 3) were performed at room temperature and dry conditions using a pre-load of 0.25 N.

### 2.6. Proliferation and Viability Assay

#### 2.6.1. Proliferation Assay

MI192-pre-treated and untreated hBMSCs were encapsulated within 5 wt% GelMA at 1 × 10^6^ cells/mL of GelMA. The cell/pre-polymer mixture was transferred into silicon moulds (Ø 5 × 2 mm), placed on a glass slide and exposed under a visible-light source (50 ± 5 mW/dm^2^) for 10 min. Once cross-linked, cell-laden hydrogels were placed into non-adherent 48-well plates and cultured in basal media. DNA content was assessed using PicoGreen per the manufacturer’s protocol (Life Technologies, Paisley, UK).

#### 2.6.2. Live/Dead Staining

At 24 h post encapsulation and after 6 weeks in osteogenic culture, the cell-laden hydrogels were incubated in a basal medium containing CellTrackerTM Green CMFDA (2 µM) (ThermoFisher Scientific, Paisley, UK) and ethidium homodimer-1 (EH-1) (4 µM) (Sigma-Aldrich, Gillingham, UK) for 45 min. The medium was replaced with a fresh medium for 30 min before being visualised under an SP8 Confocal microscope (Leica, Milton Keynes, UK) using excitation/emission filters at 488/530 nm and 530/580 nm. CellTrackerTM Green CMFDA labels viable cells as green and EH-1 labels the nuclei of dead cells as red.

### 2.7. Osteogenic Induction

MI192-pre-treated and untreated hBMSCs (5 × 10^6^ cells/mL) were encapsulated within 5 wt% GelMA. Cell/pre-polymer solution was transferred into silicon moulds (Ø 5 × 2 mm), placed on a glass slide and exposed under a visible-light source (50 ± 5 mW/dm^2^) for 10 min. Once cross-linked, cell-laden hydrogels were placed into non-adherent 48-well plates and cultured in osteogenic media for 6 weeks. The osteogenic culture media consisted of a basal medium containing 50 µM L-ascorbic acid, 10 mM β-glycerol phosphate and 100 nM dexamethasone (Sigma-Aldrich, Gillingham, UK).

### 2.8. Alkaline Phosphatase-Specific Activity (ALPSA) Assay

After 2 weeks of osteogenic culture, the cell-laden hydrogels were washed three times in PBS, then homogenised by passing through a sterile syringe. Samples were incubated with 500 µL 0.1% Triton X-100 and vortexed/sonicated for 5 min, and then samples were frozen at −80 °C. Samples were then thawed at 37 °C, and the lysing/freeze/thaw process was repeated four times. The samples were centrifuged at 10,000× *g* for 10 min at 4 °C, and lysates were collected for ALPSA assay. ALPSA quantification was determined using the 4-nitrophenyl colorimetric phosphate liquid system (pNPP, Sigma-Aldrich, Gillingham, UK) as previously described [15,34]. ALPSA was determined by normalisation with the total DNA content of the same sample. The DNA content was measured using the PicoGreen fluorescence reagent according to the manufacturer’s instructions (Life Technologies, Paisley, UK) [35].

### 2.9. Histological Analysis

The constructs were fixed in 10% neutral buffered formalin (Cellpath, Newtown, UK) for 24 h. Following fixation, samples were then embedded in paraffin wax and a microtime (Leica, Milton Keynes, UK) was used to create 4 μm sections. Picrosirius red/Alcian blue staining was undertaken to visualise collagen and glycosaminoglycans (GAGs), respectively. To quantify collagen staining, 0.5 M sodium hydroxide was used to elute the bound dye and absorbance was read at 590 nm using the Varioskan Flash Multimode Microplate Reader (Thermo Scientific, Paisley, UK). Calcium accumulation and mineralisation were assessed via Alizarin red staining (Millipore, Watford, UK) and von Kossa staining (Van Geison counterstain) (Atom Scientific, Cheshire, UK), respectively. Stained samples were captured under an Olympus BX50 microscope (Japan) and analysed using NIS Elements BR software (Ver. 3.0). Mineral nodule coverage from von Kossa-stained samples was quantified using ImageJ software. To quantify the degree of Alizarin red staining, the stained samples were eluted with 10% cetylpyridinium chloride (Sigma-Aldrich, Gillingham, UK) for 1 h, and then absorbance was measured at 550 nm using a Varioskan Flash Multimode Microplate Reader (Thermo Scientific, Paisley, UK).

### 2.10. Immunohistochemical Analysis

The degree of extracellular matrix deposition was assessed via immunohistochemical analysis using the EnVisionTM Detection Systems Peroxidase/DAB, Rabbit/Mouse (Dako, Cambridgeshire, UK) kit. Briefly, sections were incubated with ‘Dual Endogenous Enzyme Block’ from the EnVisionTM kit for 10 min before blocking with 20% normal goat serum (Dako, Cambridgeshire, UK) in PBS for 30 min. After being washed in PBS for 5 min, the primary antibodies (collagen type I: Col1a, 1:100 dilution and osteocalcin: OCN, 1:800 dilution) (Abcam, Cambridge, UK) were added to samples at the desired concentration in 1% BSA (Sigma-Aldrich, Gillingham, UK) in PBS and left to incubate overnight at 4 °C. Then, the sections were washed in PBS for 10 min, followed by incubation in the secondary HRP goat anti-rabbit IgG antibody for 30 min. After that, the Dako DAB developing solution was applied to slides for 10 min and counter stained with Harris Haematoxylin before being visualised under an Olympus BX50 microscope.

### 2.11. Statistical Analysis

All experiments were repeated at least 3 times. Data are expressed as the mean ± standard deviation (SD). All statistical analyses were conducted using ANOVA multiple comparisons test with post hoc Tukey’s test with IBM SPSS software (IBM Analytics, version 21). *p* values equal to or lower than 0.05 were considered as statistically significant. For all graphs: NS = *p* > 0.05, * *p* ≤ 0.05, ** *p* ≤ 0.01, and *** *p* ≤ 0.001.

## 3. Results

### 3.1. GelMA Hydrogels Support the Proliferation and Viability of MI192-Pre-Treated hBMSCs

The proliferation of hBMSCs within GelMA hydrogels was assessed by quantifying DNA content (Figure 1a). There was a time-dependent increase in the number of hBMSCs over 7 days of basal culture, with no significance between the MI192-pre-treated and untreated groups (*p* > 0.05). Following live/dead staining, viable cells (green) were distributed throughout the hydrogels, with very few dead cells (red) in both groups after 24 h post encapsulation (Figure 1b). Cells in both groups exhibited a spherical morphology. After 6 weeks in osteogenic culture, both MI192-pre-treated and untreated hBMSCs remained highly viable throughout the hydrogels, with little evidence of dead cells. Cellular morphology resembled a fibroblastic-like shape; however, MI192-pre-treated cells exhibited a more flattened-elongated morphology when compared to untreated cells.

### 3.2. MI192 Enhances Osteogenic Differentiation of hBMSCs within GelMA Hydrogels

ALP-specific activity, an early marker of bone formation, was assessed following 2 weeks of osteogenic culture (Figure 2). The results showed that ALPSA was significantly enhanced in MI192-pre-treated hBMSCs (1.34-fold) when compared in untreated cells (*p* ≤ 0.001).

### 3.3. MI192 Pre-Treatment Promotes the Extracellular Matrix Mineralisation of hBMSCs within GelMA Hydrogels

Picrosirius red/Alcian blue staining of constructs was conducted to visualise the production of collagen and glycosaminoglycans (GAGs), respectively (Figure 3a). Both groups displayed positive collagen deposition throughout the hydrogels, with the MI192 group showing an increased global intensity compared to the untreated group. Little staining of GAGs was observed in both groups. Quantitative analysis showed that the MI192-pre-treated groups exhibited a 1.63-fold increase in collagen production compared to the untreated control (Figure 3b) (*p* ≤ 0.001). Calcium deposition within the hydrogels was determined via Alizarin red staining. The MI192-pre-treated group exhibited enhanced calcium production throughout the hydrogels compared to that of the untreated group (Figure 3c). Quantitative analysis confirmed the significantly increased calcium deposition within the MI192-pre-treated group (1.27-fold) compared to the untreated control (Figure 3d) (*p* ≤ 0.001). Mineral nodule formation was evaluated via von Kossa staining. Both groups exhibited positive von Kossa staining (Figure 3e). However, the MI192 group showed an increase in the quantity of mineral nodules (1.37-fold) (*p* ≤ 0.001) (Figure 3f).

Extracellular matrix deposition was evaluated via immunohistochemistry. Both MI192-pre-treated and untreated hBMSCs-GelMA constructs exhibited positive staining for Col1a throughout, with increased staining intensity located at the periphery of the hydrogel (Figure 4). The MI192 group displayed increased global Col1a deposition compared to the untreated group. Both groups also showed positive OCN immunostaining, with the MI192-pre-treated constructs exhibiting substantially increased global staining for OCN compared to the control constructs (Figure 4).

### 3.4. MI192 Induced the Mineralisation of hBMSCs in PEGT/PBT-Reinforced GelMA Hydrogels

To initially evaluate the potential of improving the mechanical properties of the GelMA hydrogel via the introduction of a 3D printed structural scaffold, compressive testing was conducted. A compressive modulus of 0.0062 ± 0.0012, 6.21 ± 0.68 and 6.86 ± 0.69 MPa was acquired from the GelMA hydrogel, the PEGT/PBT scaffold and the GelMA–PEGT/PBT construct, respectively. The GelMA hydrogel samples exhibited a significantly lower modulus (~1000-fold) when compared to the other groups (*p* ≤ 0.001) (Figure 5a,b).

Picrosirius red staining (Figure 6a) showed that both MI192-pre-treated and untreated hBMSCs encapsulated within GelMA–PEGT/PBT constructs formed extensive tissue structures throughout after 6 weeks of osteogenic culture. The 3D printed scaffolds were successfully incorporated within the hydrogel system during the cross-linking process. Both groups exhibited positive picrosirius red staining for collagen production throughout the constructs, particularly at the outer regions. The MI192 construct displayed increased global collagen staining compared to the untreated group, while both groups showed little evidence of GAG expression. Quantitative analysis showed that the MI192 group elicited a 1.93-fold increase in collagen production compared to the untreated construct (*p* ≤ 0.001) (Figure 6b). Similarly, immunostaining confirmed that the MI192-pre-treated groups exhibited substantially increased global Col1a production compared to untreated constructs (Appendix A).

Calcium accumulation within the constructs was assessed via Alizarin red staining. Positive staining for calcium accumulation was observed in both MI192-pre-treated and untreated groups, with the strongest staining situated at the periphery of the constructs (Figure 6c). The MI192 group exhibited slightly stronger staining for calcium deposition when compared to the untreated group, particularly at the outer regions. Following quantitative analysis, the MI192-pre-treated group showed a 1.34-fold significant increase in calcium deposition when compared to the untreated gel (*p* ≤ 0.01) (Figure 6d).

von Kossa staining was used to evaluate the formation of functional mineral nodules. Positive staining for mineral nodules was observed in both groups, primarily located in close proximity to cellular regions of the construct (Figure 6e). The MI192-pre-treated constructs exhibited an increased density of mineral nodules when compared to the untreated group. Quantitative analysis showed a 3.38-fold enhancement in the quantity of functional mineral nodules in the MI192-pre-treated group when compared to the untreated group (*p* ≤ 0.001) (Figure 6f).

## 4. Discussion

Epigenetic reprogramming has gained increasing attention as a strategy to enhance the transcriptional activity of cells without altering the genome, therefore providing an alternative, possibly safer method of controlling MSC fate compared to genetic modification [1,36,37]. Hyperacetylation induced by HDACis has been shown to promote the osteogenic capacity of MSCs. However, there have been limited investigations regarding the efficacy of these epigenetic compounds in a 3D microenvironment. It is critical to evaluate the effectiveness of epigenetically-reprogrammed cells within a more physiological 3D environment, as cells are known to exhibit different cellular behaviour depending on the substrate they are within. Previously, we reported the enhanced osteoinductive efficacy of MI192-pre-treated hBMSCs pre-cultured as microtissues and bio-assembled within a 3D printed scaffold [33]. Although this approach shows promise, the high quantity of MSCs required to repair a critical-sized bone defect hinders its clinical utility. Hence, there is growing precedence for controlling the delivery of MSCs in situ to maximise their therapeutic efficacy. GelMA hydrogels have demonstrated their potential for different tissue engineering applications due to their plethora of desirable properties [34,38,39]. Hence, this study aimed to evaluate the capability of GelMA hydrogels reinforced with a 3D printed structural scaffold to support the MI192-induced mineralisation of hBMSCs.

HDACi-induced modifications to the epigenome have been shown to affect key cellular processes such as proliferation, differentiation and apoptosis [10]. The delivery of viable cells to the injury site is essential for the success of any MSC-based therapy targeted for bone regeneration. Thus, it is important to evaluate the effects of biomaterial systems on the viability of these epigenetically-modified cells. In our study, MI192-pre-treated hBMSCs remained highly viable 24 h post encapsulation within the hydrogels and after 6 weeks culture in osteogenic conditions, consistent with previous studies [40,41]. These results confirmed that the cross-linking procedure did not affect the viability of the epigenetically-modified cells.

Additionally, cells were observed adhering to and elongating along with the internal pore network within GelMA after the culture period, with the 5 wt% polymer concentration shown to promote cell migration and proliferation, which is essential for promoting osteogenesis [30,42]. Interestingly, MI192-pre-treated cells exhibited a more elongated morphology than untreated cells in osteoinductive culture, a morphology typically associated with a mature osteogenic phenotype [13]. To date, there have been no studies investigating the osteogenic differentiation capacity of epigenetically-modified MSCs within hydrogel systems. Therefore, to initially assess the ability of GelMA to support the mineralisation of MI192-pre-treated hBMSCs, ALPSA, an early marker of osteogenesis, was evaluated. Quantitative analysis showed a 1.34-fold increase in ALPSA, confirming the enhanced osteogenic phenotype of MI192-pre-treated hBMSCs. These results correlated with previous studies reporting increased ALPSA in MI192-pre-treated hBMSCs within the BMT construct [33]. Taken together, these results show that GelMA provides a suitable microenvironment to support the MI192-induced osteogenic differentiation of hBMSCs. Moreover, the enhanced osteogenesis observed within the GelMA hydrogel and BMT construct indicates the plasticity of this epigenetic reprogramming approach in promoting the mineralisation of hBMSCs in different 3D culture microenvironments.

The production of the extracellular matrix of MSCs and its subsequent mineralisation are essential to the functional efficacy of tissue-engineered constructs targeted for bone repair [43]. In this study, we evaluated the capacity of MI192 pre-treatment to promote the extracellular matrix mineralisation of hBMSCs within the GelMA hydrogel. Our findings showed that the MI192-pre-treated constructs exhibited a significantly enhanced extracellular matrix collagen deposition (1.63-fold) compared to the untreated cell-laden constructs. Utilising immunohistochemistry, we confirmed that the MI192-pre-treated cell-laden hydrogels displayed increased Col1a and OCN deposition, key osteoblast-related matrix components involved in mineralisation [44]. These findings are consistent with the effects of MI192 pre-treatment on the osteoblast-related protein production of hDPSCs [17]. In addition to increased intensity, Col1a and OCN deposition were distributed more uniformly throughout the hydrogels. This more homogenous distribution of osteoblast-related markers within the MI192-pre-treated hydrogels correlated with extracellular matrix deposition from the MI192-pre-treated hBMSC BMTs [33]. With the translation of MSC-based therapies hindered by the inherent heterogeneity of the cells, these results indicate that MI192-induced epigenetic activation can prime the encapsulated hBMSCs with enhanced osteogenic capacity, resulting in a more homogenous distribution of extracellular matrix proteins in the hydrogel. As the extracellular matrix provides a template for mineralisation, we assessed the effect of MI192 pre-treatment on the calcium deposition and mineral nodule formation of hBMSCs within the hydrogel. Our findings showed that the MI192-pre-treated hydrogels exhibited significantly increased calcium deposition (1.27-fold) and mineralisation (1.37-fold) when compared with the untreated group, as shown by Alizarin red and von Kossa staining, respectively. The enhanced extracellular matrix mineralisation observed within the MI192-pre-treated gels is likely due to the prolonged effects of this HDACi following treatment. Boissinot et al. reported that MI192 exhibited slow on/off binding kinetics to HDAC isoforms when compared to other HDACis [45], which likely potentiates the efficacy of MI192 pre-treatment during osteogenic culture. The enhanced mineralisation observed in this study is well correlated with the effects of MI192 pre-treatment on ADSCs on silk scaffolds [16] and hBMSCs in BMT constructs [33]. These findings provide increased evidence for the plasticity of this epigenetic reprogramming approach in promoting the bone formation capacity of MSCs derived from different tissue sources and within different 3D microenvironments, ultimately increasing the clinical utility of epigenetically-enhanced MSCs. Together, the results of this study show that MI192 pre-treatment accelerates osteogenic differentiation of hBMSCs within the GelMA hydrogel, enhancing the formation of bone-like tissue.

Although the promise of utilising GelMA hydrogels to promote bone formation has been reported, their lack of mechanical strength hinders their clinical applicability for load-bearing tissues [46]. Researchers have investigated approaches to improve the mechanical properties of GelMA hydrogels targeted for bone repair. For example, Celikkin et al. evaluated the effect of increasing polymer concentration to enhance the utility of GelMA for bone regeneration. Their findings demonstrated that 10 wt% GelMA significantly improved the compressive modulus compared to 5 wt% GelMA. However, the mineralisation of MSCs was substantially increased in the softer gel [30]. Several studies have further investigated the incorporation of additives into hydrogels as an alternative approach to promote their mechanical strength. For example, the inclusion of synthetic nanosilicates has been reported to significantly improve the mechanical properties of GelMA hydrogels [34,47,48]. However, the nanosilicate concentration required to increase GelMA material properties to that of load-bearing tissues may have a detrimental effect on cell viability [49]. Hence, retaining the favourable cellular microenvironment of low wt% GelMA, whilst increasing its mechanical strength, is critical for its use in bone augmentation strategies. To address the limitations of low wt% GelMA hydrogels for bone tissue engineering, we reinforced the hBMSC-laden GelMA hydrogel with the 3D printed PEGT/PBT scaffold. PEGT/PBT block copolymers provide greater control on printed scaffold degradation and mechanical properties when compared to polycaprolactone (PCL) [50,51].

Moreover, this specific polymer/scaffold architecture has been reported to exhibit mechanical strength similar to that of load-bearing tissues [32]. Histological analysis demonstrated that the 3D printed scaffold was successfully incorporated within the cell-laden GelMA hydrogel, indicating that it did not interfere with the photo-cross-linking of GelMA. GelMA at the interface of the 3D printed scaffold fibres was able to cross-link at a similar degree compared to GelMA in the direct path of the light (i.e., within the pores of the 3D printed scaffold). This may be due to the diffraction function of light during the cross-linking process. Importantly, we demonstrated a 1000-fold increase in the compressive modulus of GelMA–PEGT/PBT constructs compared to the hydrogel alone. This strategy to promote the mechanical properties of the hydrogel for load-bearing tissues has also been reported in the literature. Galarraga et al. increased the compressive modulus of norbornene-modified hyaluronic acid hydrogels 50-fold via incorporating PCL microfibers produced by melt-electrowriting, enhancing their potential for cartilage repair [52]. Therefore, these findings demonstrate the importance of incorporating 3D printed PEGT/PBT scaffolds in improving the clinical utility of GelMA hydrogels targeted for bone repair.

Although the incorporation of 3D printed PEGT/PBT scaffolds significantly enhanced the compressive modulus of GelMA hydrogels, it remained unclear how the inclusion of an external scaffold framework would impact MI192-induced osteogenesis within the composite construct. Hence, the capacity of GelMA–PEGT/PBT constructs to support the mineralisation of MI192-induced hBMSCs was evaluated. Picrosirius red staining confirmed collagen production within the GelMA–PEGT/PBT construct in both groups. MI192-pre-treated cells displayed a 1.93-fold enhancement in collagen production compared to the untreated group. In addition to extracellular collagen deposition, calcium accumulation and mineral nodule formation were evaluated. Both groups elicited substantial calcium deposition throughout the construct, with the MI192-pre-treated composite exhibiting a 1.34-fold increase in calcium accumulation compared to the untreated construct. Similarly, following assessment of mineral nodule formation via von Kossa staining, the MI192 group displayed an increased quantity of functional mineral nodules (3.38-fold) compared to the untreated group, consistent with the effects of MI192 on hBMSCs within the GelMA hydrogel alone and the BMT construct [33]. Together, these findings indicate that MI192 pre-treatment is capable of accelerating the capacity of all encapsulated hBMSCs to differentiate into a more mature osteogenic phenotype, resulting in enhanced global extracellular matrix collagen deposition and mineralisation.

Interestingly, MI192-pre-treated hBMSCs elicited increased collagen deposition and mineralisation within the GelMA–PEGT/PBT construct when compared to the GelMA hydrogel alone (picrosirius red = 1.93-fold vs. 1.63-fold, Alizarin red = 1.34-fold vs. 1.27-fold, and von Kossa = 3.38-fold vs. 1.37-fold, respectively). This enhanced bone-like tissue formation could be due to the effect of the incorporated 3D printed scaffold on imparting mechanotransductive stimuli to the encapsulated epigenetically-primed cells [53,54]. Several studies have reported the impact of biomaterial substrates in augmenting the epigenetic functionality of cells, ultimately promoting their differentiation capacity [55,56]. For example, it was reported that 3D printed titanium scaffolds with a triangle pore shape significantly enhanced osteoblast mineralisation through hyperacetylation-induced osteogenic gene activation compared to cells on square pored scaffolds [57]. Thus, it is likely that the introduction of the much stiffer PEGT/PBT scaffold within the GelMA hydrogel further potentiated the transcriptional permissiveness of the encapsulated epigenetically-primed hBMSCs through mechano-epigenetic regulation, although this would require further investigation. Moreover, the hydrogel and 3D printed scaffold interface may facilitate nutrient/waste exchange throughout the hydrogel, improving the differentiation of MI192-pre-treated hBMSCs. These findings indicate that GelMA–PEGT/PBT constructs provide a suitable platform to support the mineralisation of MI192-induced hBMSCs.

Taken together, the findings in this study demonstrate the considerable potential of harnessing the GelMA–PEGT/PBT construct to facilitate the delivery of epigenetically-primed MSCs for bone augmentation strategies. Future studies have a tremendous scope to investigate the effects of bioactive 3D printed structural scaffolds with different architectures that may further facilitate MI192-pre-treated MSCs for bone formation within the constructs. Future work will implement these GelMA–PEGT/PBT constructs to repair critical-sized bone defects in vivo.

## 5. Conclusions

In conclusion, these findings demonstrate the capacity of GelMA hydrogels reinforced with a 3D printed PEGT/PBT scaffold to support the osteogenic capacity of MI192-pre-treated hBMSCs, whilst increasing their mechanical strength, indicating the potential of this composite biomaterial to facilitate the delivery of epigenetically-primed MSCs for bone regeneration.

## Figures and Tables

**Figure 1 jfb-13-00041-f001:**
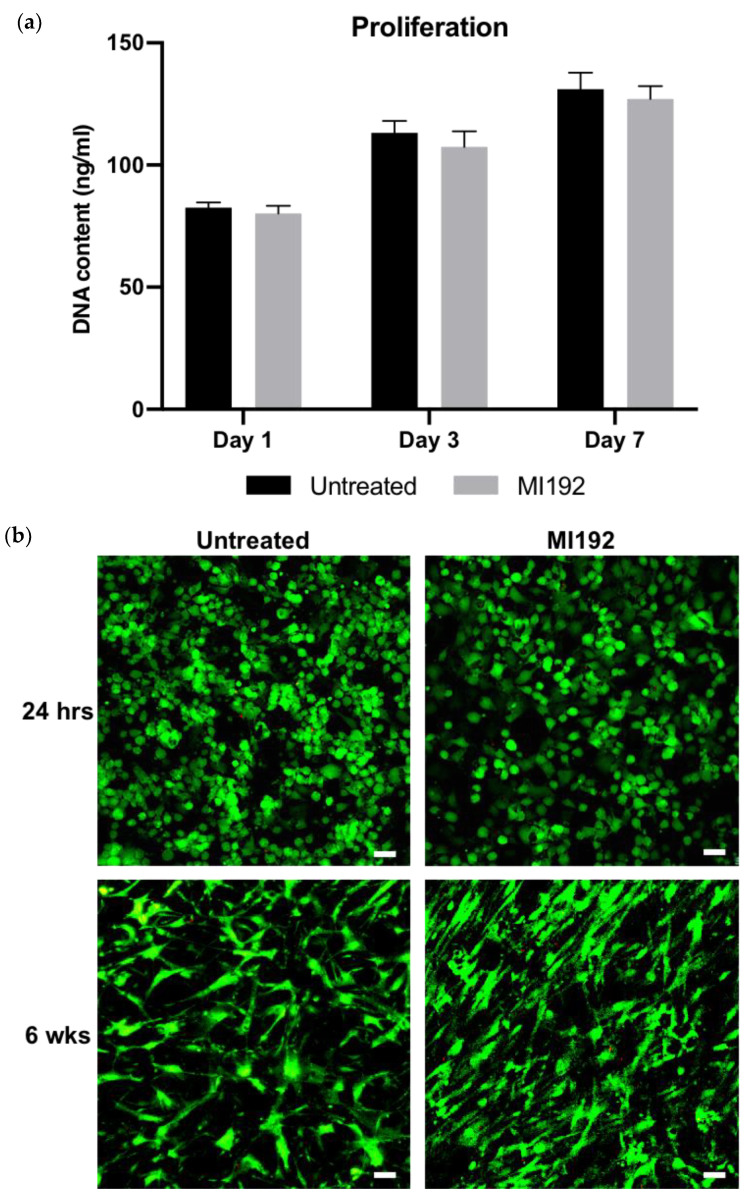
The effects of MI192 on the proliferation and viability of hBMSCs within GelMA hydrogels. (**a**) DNA content of MI192-pre-treated/untreated hBMSCs within GelMA hydrogels during basal culture. (**b**) Merged live/dead staining of encapsulated hBMSCs within GelMA during osteogenic culture (live cells—green; dead cells—red). Scale bars = 50 µm. Data are presented as the mean ± SD (*n* = 3).

**Figure 2 jfb-13-00041-f002:**
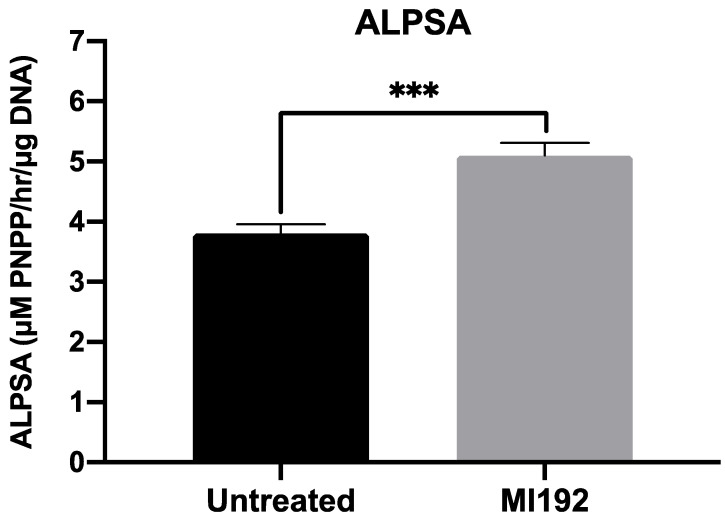
MI192 pre-treatment (48 h) enhances ALPSA in hBMSCs encapsulated in GelMA hydrogels compared with the untreated group after 2 weeks of osteogenic culture. Data are presented as the mean ± SD (*n* = 3). *** *p* ≤ 0.001.

**Figure 3 jfb-13-00041-f003:**
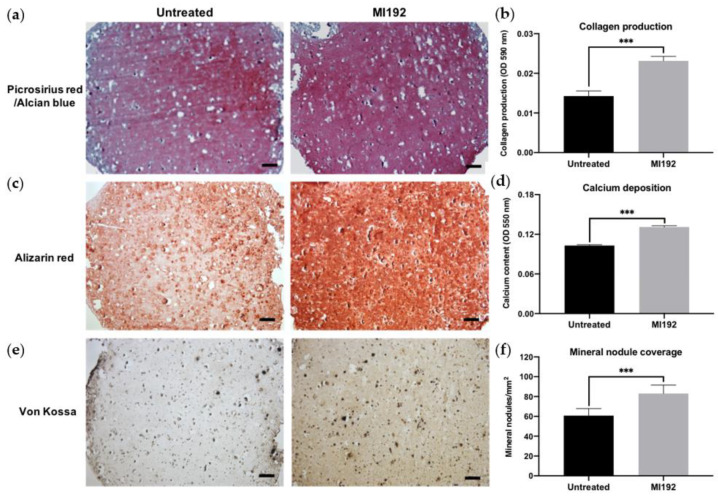
Histological staining of MI192-pre-treated hBMSCs encapsulated within GelMA hydrogels after 6 weeks of osteogenic culture. (**a**) Picrosirius red staining for collagen production in the MI192-pre-treated constructs. (**b**) Quantitative analysis of picrosirius red collagen staining. (**c**) Alizarin red staining for calcium deposition with MI192-pre-treated hBMSCs. (**d**) Quantitative analysis of Alizarin red staining. (**e**) von Kossa staining for mineral nodules. (**f**) Semi-quantitative analysis of mineral nodules. Scale bars = 100 µm. Data are presented as the mean ± SD. *** *p* ≤ 0.001.

**Figure 4 jfb-13-00041-f004:**
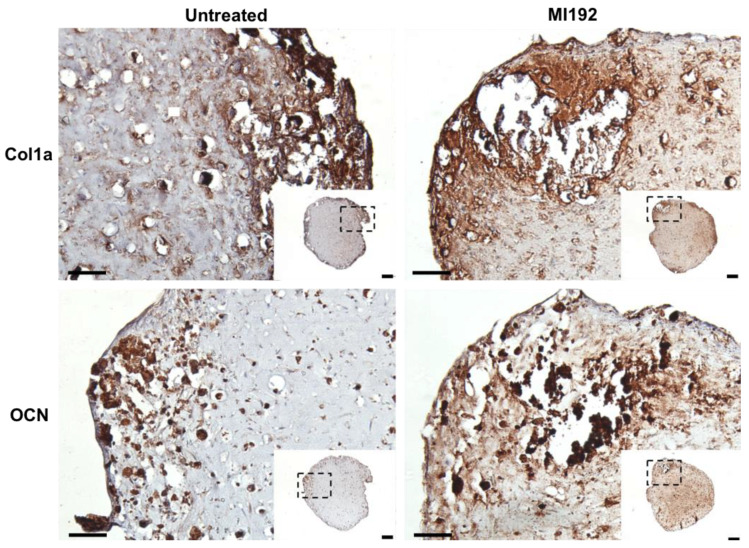
Effects of MI192 pre-treatment on Col1a and OCN expression in hBMSCs within GelMA hydrogels after 6 weeks of osteogenic culture. Scale bars = 200 and 50 µm for the low and high magnification, respectively.

**Figure 5 jfb-13-00041-f005:**
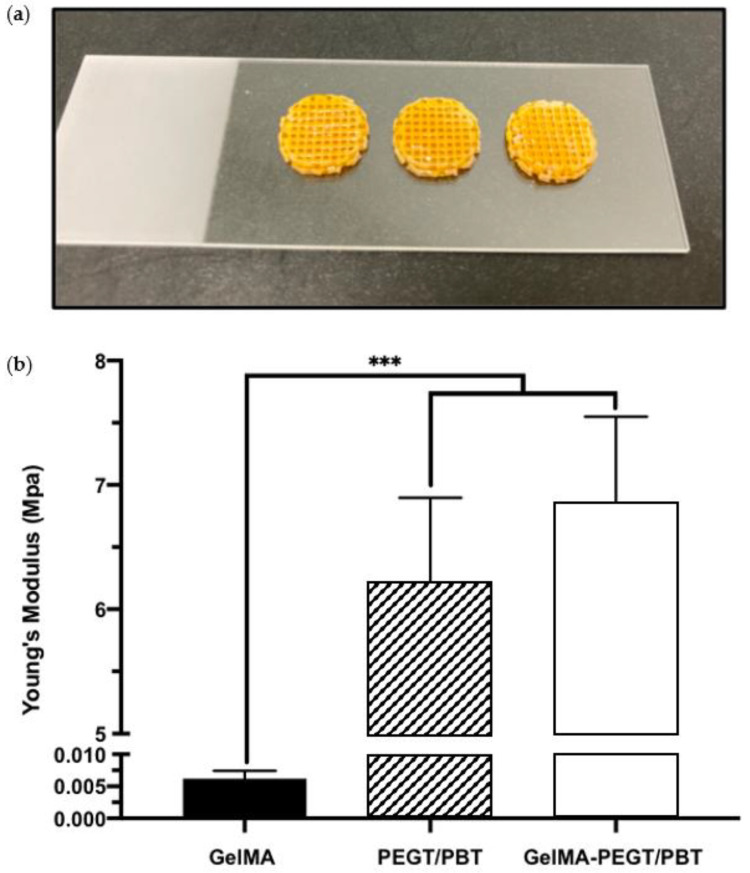
Mechanical properties of GelMA, PEGT/PBT and the GelMA–PEGT/PBT construct. (**a**) Representative images of the GelMA–PEGT/PBT construct following photo-curing. (**b**) Compressive modulus of the GelMA hydrogel, the PEGT/PBT scaffold and the GelMA–PEGT/PBT construct. Data are presented as the mean ± SD. *** *p* ≤ 0.001.

**Figure 6 jfb-13-00041-f006:**
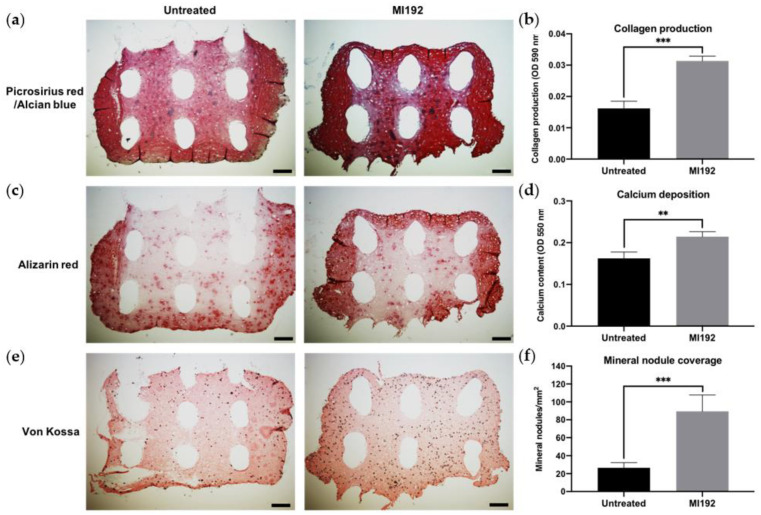
Histological staining on the cross-sections of MI192 pre-treated/untreated hBMSCs encapsulated within GelMA–PEGT/PBT constructs after 6 weeks of osteogenic culture. (**a**) Picrosirius red staining for collagen production in the MI192-pre-treated GelMA–PEGT/PBT constructs. (**b**) Quantitative analysis of picrosirius red collagen staining. (**c**) Alizarin red staining for calcium deposition with the MI192-pre-treated GelMA–PEGT/PBT constructs. (**d**) Quantitative analysis of Alizarin red staining. (**e**) von Kossa staining for mineral nodules. (**f**) Semi-quantitative analysis of mineral nodules. Scale bars = 200 µm. Data are presented as the mean ± SD. ** *p* ≤ 0.01 and *** *p* ≤ 0.001. Note: the white elliptical spaces within the stained sections are the 3D printed PEGT/PBT fibres.

## Data Availability

Not applicable.

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
