# Peer review of "GelMA Hydrogel Reinforced with 3D Printed PEGT/PBT Scaffolds for Supporting Epigenetically-Activated Human Bone Marrow Stromal Cells for Bone Repair"

_jfb, 2022, doi:10.3390/jfb13020041_

Round 1
Reviewer 1 Report
This study aimed to investigate a GelMA hydrogel reinforced with a 3D printed scaffold to support MI192-induced human bone marrow stromal cells (hBMSCs) for bone formation.
This study conducted MI192 pretreatment and showed effective results.
In addition, this study proposes an effective method of making bone-forming scaffolds by printing GelMA-PEGT/PBT.
Author Response
Dear Reviewers,
On behalf of all the authors, I would like to thank you for your valuable feedback and suggestions to improve the quality of our manuscript. We have carefully revised our manuscript to address all of these comments and suggestions and included a point by point response below in red text with the changes highlighted in the manuscript. The English grammar has also been checked by native English speaker.
We hope the revised manuscript will now be acceptable.
Yours sincerely,
Xuebin
Reviewer 1
This study aimed to investigate a GelMA hydrogel reinforced with a 3D printed scaffold to support MI192-induced human bone marrow stromal cells (hBMSCs) for bone formation.
This study conducted MI192 pretreatment and showed effective results.
In addition, this study proposes an effective method of making bone-forming scaffolds by printing GelMA-PEGT/PBT.
Answer: We would like to thank the reviewer for the very positive comments and his/her interesting in our work.
Reviewer 2 Report
The manuscript by Man et al. evaluated the activities of human bone marrow stromal cells using GelMA hydrogels and PEGT/PBT-GelMA constructs. There are major concerns in the manuscript.
- In section 2.2 and 2.3, both GelMA hydrogels and PEGT/PBT-GelMA constructs were produced by photo crosslinking.
(a) How does 3D printed PEGT/PBT scaffolds play a role in photo crosslinking of GelMA? For example, it is expected that portion of the light will be blocked by the PEGT/PBT scaffolds, resulting in portions of uncrosslinked GelMA.
(b) The authors should compare cell activities before and after photo crosslinking for both of the tissue scaffolds, especially when the hydrogels contained photo initiator. - The control groups are missing for all the cell studies, meaning the cell activities using only tissue culture plates.
- Histological images in Figure 6 are particular confusing. It appears that the authors removed the tissue constructs and evaluated the level of collagen production, calcium deposition, and mineral nodule coverage on tissues outside the tissue constructs. Does this mean that the authors are suggesting the tissue constructs promote cellular activities outside the scaffold?
Author Response
Dear Reviewers,
On behalf of all the authors, I would like to thank you for your valuable feedback and suggestions to improve the quality of our manuscript. We have carefully revised our manuscript to address all of these comments and suggestions and included a point by point response below in red text with the changes highlighted in the manuscript. The English grammar has also been checked by native English speaker.
We hope the revised manuscript will now be acceptable.
Yours sincerely,
Xuebin
Reviewer 2
The manuscript by Man et al. evaluated the activities of human bone marrow stromal cells using GelMA hydrogels and PEGT/PBT-GelMA constructs. There are major concerns in the manuscript.
1. In section 2.2 and 2.3, both GelMA hydrogels and PEGT/PBT-GelMA constructs were produced by photo crosslinking.
a) How does 3D printed PEGT/PBT scaffolds play a role in photo crosslinking of GelMA? For example, it is expected that portion of the light will be blocked by the PEGT/PBT scaffolds, resulting in portions of uncrosslinked GelMA.
Answer: Thank you for your questions. We did not observed notable evidence of any un-cross-linked GelMA in the histology analysis of our constructs (Figure 6 and S1), where the hydrogel formed a tight boundary around the 3D printed fibres. This indicates the GelMA at the interface of the 3D printed scaffold fibres was able to crosslink at the similar degree compared to the GelMA in the direct path of the light (i.e. within the pores of the 3D printed scaffold). This may be due to the diffraction function of light during the crosslink. Moreover, Cesar et al (2022) reported the successful crosslinking of the same formulation of GelMA/photo initiators when combined with a 3D printed PCL scaffold [1], consistent with the observations in our study. Together, these findings indicating the high crosslinking efficiency of the GelMA/photo initiator systems used in this study.
1) Cesar R. Alcala-Orozco, Isha Mutreja, Xiaolin Cui, Gary J. Hooper, Khoon S. Lim, Tim B.F. Woodfield, Hybrid biofabrication of 3D osteoconductive constructs comprising Mg-based nanocomposites and cell-laden bioinks for bone repair, Bone, 2022, 154,116198,
(b) The authors should compare cell activities before and after photo crosslinking for both of the tissue scaffolds, especially when the hydrogels contained photo initiator.
Answer: Thank you for your suggestion. As GelMA hydrogels using this formulation of visible light photo initiator has shown its biocompatible nature in this study (Figure 1) and has been extensively reported in the literature, showing its biocompatibility for different cells/clinical applications [1-3], the cell activities within the uncured pre-polymer was not evaluated. Moreover, the main aim of this study was to determine the effects of epigenetic activation of the cell behaviour within the fully cross-linked construct. Having observed a similar degree of osteogenic enhancement due to epigenetic priming, suggests the limited exposure within the uncured pro-polymer solution did not adversely impact the biological function of the cells.
1) Man, K.B., I. A.; Brunet, M. Y.; Federici, A. S.; Peacock, B.; Hoey, D. A.: and Cox, S. C. Controlled release of epigenetically-enhanced extracellular vesicles from a gelma/nanoclay composite hydrogel to promote bone repair. The International Journal of Molecular Sciences 2022, 23, 832.
2) Mekhileri, N.V.; Lim, K.S.; Brown, G.C.J.; Mutreja, I.; Schon, B.S.; Hooper, G.J.; Woodfield, T.B.F. Automated 3d bioassembly of micro-tissues for biofabrication of hybrid tissue engineered constructs. Biofabrication 2018, 10, Artn 024103.
3) Tang, J., Cui, X., Zhang, Z., Xu, Y., Guo, J., Soliman, B. G., Lu, Y., Qin, Z., Wang, Q., Zhang, H., Lim, K. S., Woodfield, T. B F., Zhang, J., Injection-Free Delivery of MSC-Derived Extracellular Vesicles for Myocardial Infarction Therapeutics. Adv. Healthcare Mater. 2022, 11, 2100312.
2. The control groups are missing for all the cell studies, meaning the cell activities using only tissue culture plates.
Answer: As we have previously established the role of MI192 on stimulating hBMSCs osteogenesis in 2D culture, the primary aim of this study was to determine the influence of the GelMA constructs and/or with 3D printed scaffolds in supporting epigenetically-activated cells osteogenesis to support MI192-induced bone formation. Therefore, the untreated cells were used as the controls.
3. Histological images in Figure 6 are particular confusing. It appears that the authors removed the tissue constructs and evaluated the level of collagen production, calcium deposition, and mineral nodule coverage on tissues outside the tissue constructs. Does this mean that the authors are suggesting the tissue constructs promote cellular activities outside the scaffold?
Answer: To clarify, all the histological stains are within the tissue constructs. Initially, the cells were encapsulated with the 5 wt% GelMA hydrogel. For the GelMA-PEGT/PBT constructs, the PEGT/PBT scaffolds were inserted into the cell/pre-polymer solution prior to cross-linking. The white gaps within the sections are where the 3D printed scaffold fibres would have been, however, they were displaced from the sections during the histological processing of the samples, resulting in the empty voids seen in the images. It is more probable the increased staining for extracellular matrix collagen production and mineralisation observed at the outer regions of the construct is likely due to the increased proximity of cells in these locations and the diffusion mechanism in the culture medium. This suggests that ‘the tissue constructs promote cellular activities at the outer regions of the construct’, but not ‘outside the scaffold’.
Reviewer 3 Report
The work describes the use of 3D-printed scaffolds to reinforce the mechanical strength of a hydrogel matrix used for bone repair. As a result, the production of extracellular matrix collagen mineralization was enhanced. Bioprinting thus seems a novel, promising strategy to promote the osteogenic capacity of epigenetically reprogrammed cells.
The work is well written, organized and discussed, with minor flaws, which need to be addressed and/or clarified:
The use of too many abbreviations in the abstract makes reading difficult and should be avoided, particularly when they are unnecessary since they are never used again in the abstract (e.g. HDAC L11); others are not presented in full the first they appear (e.g. Col1a and OCN, L20).
- Notation of degrees centigrade should be corrected throughout the text to ‘°C’.
- Polymers do not dissolve but rather disperse in water. Therefore, (e.g. L92-93) it should read: ‘until fully dispersed’ and ‘gelatin dispersion’. Please throughout the text, whenever applicable.
L94-95 – Extensive dialysis at 40°C is bound to have resulted in protein degradation. Salt and methacrylic acid removal could have been performed by a more expedite and conservative method (e.g. centrifugation of dialysis tubes). Please comment.
L110-111 – The 3D printing process should be more detailed and understandable by readers unfamiliar with it, namely referring to the type of technology and processing parameters used.
L113-114 – The use of 70% ethanol as a disinfectant is expected to reduce microbiological burden but achievement of sterility, as mentioned, in not guaranteed. How did you ascertain that a sterility level (10-6 CFU) was achieved?
L129 – Country of the mechanical testing machine and the number of measurements should be disclosed.
L156-157 – What is meant by ‘three times in 1x PBS’?
L200 – The usefulness of Table 1 is unclear; it seems that the relevant details of the antibodies could have been given in the text. Additionally, is the catalog number relevant?
L500 - Suppliment material?
Captions/Legends to Figures have a font different from the text; please use the same font and size.
Author Response
Dear Reviewers,
On behalf of all the authors, I would like to thank you for your valuable feedback and suggestions to improve the quality of our manuscript. We have carefully revised our manuscript to address all of these comments and suggestions and included a point by point response below in red text with the changes highlighted in the manuscript. The English grammar has also been checked by native English speaker.
We hope the revised manuscript will now be acceptable.
Yours sincerely,
Xuebin
Reviewer 3
The work describes the use of 3D-printed scaffolds to reinforce the mechanical strength of a hydrogel matrix used for bone repair. As a result, the production of extracellular matrix collagen mineralization was enhanced. Bioprinting thus seems a novel, promising strategy to promote the osteogenic capacity of epigenetically reprogrammed cells.
Answers: Thanks for the positive comment and recognise the novelty of our work.
The work is well written, organized and discussed, with minor flaws, which need to be addressed and/or clarified:
Answer: We were pleased to see that the reviewer found our manuscript interesting and, in this revision, we have addressed his/her comments in full (see below).
The use of too many abbreviations in the abstract makes reading difficult and should be avoided, particularly when they are unnecessary since they are never used again in the abstract (e.g. HDAC L11); others are not presented in full the first they appear (e.g. Col1a and OCN, L20).
Answer: Thanks. As suggested, we have removed the abbreviations of HDAC, Col1a and OCN in the abstract.
- Notation of degrees centigrade should be corrected throughout the text to ‘°C’.
Answer: Thanks for picking up this, we have amended this mistake throughout the manuscript.
- Polymers do not dissolve but rather disperse in water. Therefore, (e.g. L92-93) it should read: ‘until fully dispersed’ and ‘gelatin dispersion’. Please throughout the text, whenever applicable.
Answer: Thank you for highlighting this, we have removed the term dissolved in this section and replaced this as suggested by the reviewer (first paragraph on page 3).
L94-95 – Extensive dialysis at 40°C is bound to have resulted in protein degradation. Salt and methacrylic acid removal could have been performed by a more expedite and conservative method (e.g. centrifugation of dialysis tubes). Please comment.
Answer: Thank you for your question. As gelatin is already denatured to start with being a product of collagen hydrolysation, we are not too concerned about the molecular weights of the gelatin as our experiments are based on wt% and degree of crosslinking, and it has been previously shown as a sufficient method to control the physico-chemical and mechanical properties of the gels [1,2], which is adequate for our study. Moreover, the protocol utilised for the fabrication of GelMA in this study has been well established in the literature [3].
1) Cidonio, G.; Alcala-Orozco, C.R.; Lim, K.S.; Glinka, M.; Mutreja, I.; Kim, Y.H.; Dawson, J.I.; Woodfield, T.B.F.; Oreffo, R.O.C. Osteogenic and angiogenic tissue formation in high fidelity nanocomposite laponite-gelatin bioinks. Biofabrication 2019, 11, ARTN 035027.
2) Yoon HJ, Shin SR, Cha JM, Lee SH, Kim JH, et al. (2016) Cold Water Fish Gelatin Methacryloyl Hydrogel for Tissue Engineering Application. PLOS ONE 11(10): e0163902
3) Loessner, D., Meinert, C., Kaemmerer, E. et al. Functionalization, preparation and use of cell-laden gelatin methacryloyl–based hydrogels as modular tissue culture platforms. Nat Protoc 11, 727–746 (2016).
L110-111 – The 3D printing process should be more detailed and understandable by readers unfamiliar with it, namely referring to the type of technology and processing parameters used.
Answer: We agree with the reviewer that it will be beneficial to provide more details for readers unfamiliar with it. Therefore, we have implemented a greater level of detail regarding the printing parameters in this section.
L113-114 – The use of 70% ethanol as a disinfectant is expected to reduce microbiological burden but achievement of sterility, as mentioned, in not guaranteed. How did you ascertain that a sterility level (10-6 CFU) was achieved?
Answer: Thank you for the comment. It is well know that ‘70% ethanol is commonly used as an antiseptic in lab environments. It has the highest effective concentration at 70% compared to stronger solutions’ [1]. Clinically, the surgical instruments can be sterile within 70% ethanol for 30mins for emergency use. Therefore, we use 70% ethanol overnight should be sufficient for our purpose in the laboratory. This was to ensure that the sterilisation procedure did not differentially impact the biological/mechanical properties of these 3D printed scaffolds, standardising the reproducible fabrication/processing of the scaffolds across different studies. We did not observe any microbial contamination in our work using this disinfectant protocol. But we do agree with the reviewer that the sterility of the scaffold cannot be guaranteed for clinical procedure, which is beyond the scope of this present study.
1) https://www.laballey.com/collections/ethanol-140-proof#:~:text=70%25%20ethanol%20is%20commonly%20used%20as%20an%20antiseptic,and%20equipment%20surfaces%20in%20labs%20and%20production%20facilities.
L129 – Country of the mechanical testing machine and the number of measurements should be disclosed.
Answer: Thank you for pointing out this exclusion. We have now implemented the information regarding the country and the number of measurements made.
L156-157 – What is meant by ‘three times in 1x PBS’?
Answer: This is meant to describe the washing of samples with PBS at 1 x concentration for three times. To provide better clarity, the “1 x” has now been removed.
L200 – The usefulness of Table 1 is unclear; it seems that the relevant details of the antibodies could have been given in the text. Additionally, is the catalog number relevant?
Answer: we agree with the reviewer that it is better to add the table content in the text. As suggested the primary antibodies have been added in the main text and the table has been removed.
L500 - Suppliment material?
Answer: Sorry for the typo. This has now been corrected to read as “Supplementary material”
Captions/Legends to Figures have a font different from the text; please use the same font and size.
Answer: We apologies that the Figure captions in Figure S1 are not consistent with the rest of the figures in the manuscript, this has now been corrected.
Round 2
Reviewer 2 Report
The authors have explained the deficiencies from the reviewer. The only concern is that the authors did no address these questions in the manuscript. Please incorporate the responses in the manuscript.
Author Response
Reviewer 2
The authors have explained the deficiencies from the reviewer. The only concern is that the authors did no address these questions in the manuscript. Please incorporate the responses in the manuscript.
Answer: Thanks for the reviewer's request. Please see our responses in 'blue' colour. We have addressed the questions in the manuscript. accordingly.
1. In section 2.2 and 2.3, both GelMA hydrogels and PEGT/PBT-GelMA constructs were produced by photo crosslinking.
a) How does 3D printed PEGT/PBT scaffolds play a role in photo crosslinking of GelMA? For example, it is expected that portion of the light will be blocked by the PEGT/PBT scaffolds, resulting in portions of uncrosslinked GelMA.
Answer: Thank you for your questions. We did not observed notable evidence of any un-cross-linked GelMA in the histology analysis of our constructs (Figure 6 and S1), where the hydrogel formed a tight boundary around the 3D printed fibres. This indicates the GelMA at the interface of the 3D printed scaffold fibres was able to crosslink at the similar degree compared to the GelMA in the direct path of the light (i.e. within the pores of the 3D printed scaffold). This may be due to the diffraction function of light during the crosslink. Moreover, Cesar et al (2022) reported the successful crosslinking of the same formulation of GelMA/photo initiators when combined with a 3D printed PCL scaffold [1], consistent with the observations in our study. Together, these findings indicating the high crosslinking efficiency of the GelMA/photo initiator systems used in this study.
To reflect this, the following sentences have been added in the discussion (Page 13, line 441-444):
The GelMA at the interface of the 3D printed scaffold fibres was able to crosslink at the similar degree compared to the GelMA in the direct path of the light (i.e. within the pores of the 3D printed scaffold). This may be due to the diffraction function of light during the crosslink.
1) Cesar R. Alcala-Orozco, Isha Mutreja, Xiaolin Cui, Gary J. Hooper, Khoon S. Lim, Tim B.F. Woodfield, Hybrid biofabrication of 3D osteoconductive constructs comprising Mg-based nanocomposites and cell-laden bioinks for bone repair, Bone, 2022, 154,116198,
(b) The authors should compare cell activities before and after photo crosslinking for both of the tissue scaffolds, especially when the hydrogels contained photo initiator.
Answer: Thank you for your suggestion. As GelMA hydrogels using this formulation of visible light photo initiator has shown its biocompatible nature in this study (Figure 1) and has been extensively reported in the literature, showing its biocompatibility for different cells/clinical applications [1-3], the cell activities within the uncured pre-polymer was not evaluated. Moreover, the main aim of this study was to determine the effects of epigenetic activation of the cell behaviour within the fully cross-linked construct. Having observed a similar degree of osteogenic enhancement due to epigenetic priming, suggests the limited exposure within the uncured pro-polymer solution did not adversely impact the biological function of the cells.
These can be seen in the discussion, the last sentence on line 360-262 (cell viability) and line 375-380 (osteogenesis) on page 12.
1) Man, K.B., I. A.; Brunet, M. Y.; Federici, A. S.; Peacock, B.; Hoey, D. A.: and Cox, S. C. Controlled release of epigenetically-enhanced extracellular vesicles from a gelma/nanoclay composite hydrogel to promote bone repair. The International Journal of Molecular Sciences 2022, 23, 832.
2) Mekhileri, N.V.; Lim, K.S.; Brown, G.C.J.; Mutreja, I.; Schon, B.S.; Hooper, G.J.; Woodfield, T.B.F. Automated 3d bioassembly of micro-tissues for biofabrication of hybrid tissue engineered constructs. Biofabrication 2018, 10, Artn 024103.
3) Tang, J., Cui, X., Zhang, Z., Xu, Y., Guo, J., Soliman, B. G., Lu, Y., Qin, Z., Wang, Q., Zhang, H., Lim, K. S., Woodfield, T. B F., Zhang, J., Injection-Free Delivery of MSC-Derived Extracellular Vesicles for Myocardial Infarction Therapeutics. Adv. Healthcare Mater. 2022, 11, 2100312.
2. The control groups are missing for all the cell studies, meaning the cell activities using only tissue culture plates.
Answer: As we have previously established the role of MI192 on stimulating hBMSCs osteogenesis in 2D culture, the primary aim of this study was to determine the influence of the GelMA constructs and/or with 3D printed scaffolds in supporting epigenetically-activated cells osteogenesis to support MI192-induced bone formation. Therefore, the untreated cells were used as the controls.
3. Histological images in Figure 6 are particular confusing. It appears that the authors removed the tissue constructs and evaluated the level of collagen production, calcium deposition, and mineral nodule coverage on tissues outside the tissue constructs. Does this mean that the authors are suggesting the tissue constructs promote cellular activities outside the scaffold?
Answer: To clarify, all the histological stains are within the tissue constructs. Initially, the cells were encapsulated with the 5 wt% GelMA hydrogel. For the GelMA-PEGT/PBT constructs, the PEGT/PBT scaffolds were inserted into the cell/pre-polymer solution prior to cross-linking. The white gaps within the sections are where the 3D printed scaffold fibres would have been, however, they were displaced from the sections during the histological processing of the samples, resulting in the empty voids seen in the images. It is more probable the increased staining for extracellular matrix collagen production and mineralisation observed at the outer regions of the construct is likely due to the increased proximity of cells in these locations and the diffusion mechanism in the culture medium. This suggests that ‘the tissue constructs promote cellular activities at the outer regions of the construct’, but not ‘outside the scaffold’.
To avoid confusion, ‘on the cross-sections’ has been added to the figure title and ‘Note: The white elliptical spaces within the stained sections are the 3D printed PEGT/PBT fibres’ was added in the legend.